# The Relationship between Self-Perceived Health and Clinical Symptoms in Patients with Frozen Shoulders

**DOI:** 10.3390/ijerph192114396

**Published:** 2022-11-03

**Authors:** Liang-Chien She, Hui-Yi Wang, Mei-Fang Liu, Yen-Ko Lin, Shu-Mei Chen

**Affiliations:** 1Department of Physical Medicine and Rehabilitation, Kaohsiung Medical University Hospital, Kaohsiung Medical University, Kaohsiung 807377, Taiwan; 2Department of Physical Therapy, College of Health Science, Kaohsiung Medical University, Kaohsiung 807378, Taiwan; 3Department of Medical Research, Kaohsiung Medical University Hospital, Kaohsiung Medical University, Kaohsiung 807378, Taiwan; 4Division of Trauma and Surgical Critical Care, Department of Surgery, Kaohsiung Medical University Hospital, Kaohsiung Medical University, Kaohsiung 807377, Taiwan; 5Department of Emergency Medicine, College of Medicine, Kaohsiung Medical University, Kaohsiung 807378, Taiwan; 6Department of Medical Humanities and Education, College of Medicine, Kaohsiung Medical University, Kaohsiung 807378, Taiwan

**Keywords:** frozen shoulder, pain, range of motion, self-perceived health

## Abstract

Current healthcare is centered on the perception of people’s health. The purpose of this study was to investigate the relationship between self-perceived health (physical, psychological, social, and environmental dimensions) and two main clinical symptoms (shoulder pain and restricted shoulder motion) in patients with frozen shoulders. A total of 49 patients diagnosed with frozen shoulders were recruited and divided into high- and low-disability groups according to the severity of their frozen shoulders. Participants were measured for shoulder passive range of motion, pain intensity, and self-perceived health, using a brief version of the World Health Organization Quality of Life questionnaire. The results showed that the high-disability group had poorer self-perceived health (lower quality of life scores) than the low-disability group (*p* < 0.05). There was no significant correlation between the quality of life scores and the two clinical symptoms in either the high- or low-disability group. Our findings revealed that the multidimensional self-perceived health of frozen shoulder patients could not be inferred from the severity of shoulder pain and restricted shoulder motions. This study suggests that healthcare providers should pay more attention to patients’ self-perceived health needs while addressing the clinical symptoms in patients with frozen shoulders.

## 1. Introduction

Frozen shoulder (FS) is a common condition that causes chronic shoulder problems due to inflammation and adhesions of the shoulder joint capsule, also known as adhesive capsulitis [1]. Frozen shoulder (FS) is a common chronic shoulder disorder with an estimated prevalence of at least 2% in the general population, with middle-aged people, females, and the non-dominant shoulder being the most affected [1,2,3]. Patients with FS often suffer from shoulder pain and range of motion (ROM) limitations as the main clinical symptoms; moreover, the course of the disease is often long-term [4,5,6], which leads to further functional disability that affects their quality of life [7,8]. To understand the severity of such disability due to the two main clinical symptoms in FS, an assessment tool, the Disability of the Arm, Shoulder and Hand (DASH) questionnaire, has been validated as suitable for assessing the severity of the FS. This questionnaire, including the Taiwan version, is often used for clinical studies and has good reliability and validity [8,9,10,11]. Current medical strategies for patients with FS, including physical therapy, various invasive therapies, and medications, focus on pain relief and increasing shoulder motion, with the assumption that they will help patients regain their health, i.e., improve health-related quality of life [12,13].

Patients with FS have perceived their health status through the activities of daily living, including experiencing shoulder pain and shoulder ROM limitations while performing certain physical activities (e.g., raising hands overhead, combing hair, or washing the back) [14,15,16]. Those physical limitations can cause psychological anxiety or depression [17,18], which could reduce their willingness to participate in social activities and could even result in encounters with environmental facilities that are unfavorable to them (e.g., having difficulty pulling the overhead handrail when taking public transportation) [14,15]. Accordingly, FS patients’ perceptions of their health status, as experienced in four dimensions (physical, psychological, social, and environmental interactions), can be expressed in terms of health-related quality of life. Since current healthcare policy is patient-centered, the main value is in understanding the subjective health perceptions of patients with disabilities, which is essential for treatment planning decisions.

Evidence has shown that current treatment strategies may be insufficient to satisfy the healthcare needs of patients with FS [13,15,19,20]. Patients with FS often have different severity of shoulder pain and shoulder ROM limitations, but healthcare professionals often address only those clinical symptoms, ignoring the impacts of the patient’s self-perception of health [14,15]. It is important to be aware of whether the patient’s subjective perception of health is consistent with the measured shoulder pain intensity and shoulder ROM. 

The impacts of two clinical symptoms (shoulder ROM limitations and pain) on self-perceived health in patients with FS are multidimensional; however, few studies have investigated the association between the dimensions of patients’ self-perceived health and those clinical symptoms, especially in patients of varying severity. Understanding the interactions between the patient’s self-perceived health and the two clinical symptoms in FS will provide valuable insights for clinical decision-making in line with the patient-centered perspective of healthcare. Therefore, the purpose of this study was to investigate the relationship between self-perceived health (including physical, psychological, social, and environmental dimensions) and the two clinical symptoms in patients with FS.

## 2. Materials and Methods

This prospective cross-sectional study recruited outpatients undergoing physical therapy for shoulder problems from a local hospital center in southern Taiwan. The study was conducted from 1 January 2017 to 31 May 2018. The study protocol was approved by the Ethics Committee of the University Hospital (KMUHIRB-E(II)-20160152).

### 2.1. Participants

This study included patients diagnosed with unilateral FS or adhesive capsulitis by their physicians, aged 30–70 years, with shoulder pain for more than 4 weeks and marked limitations in both active and passive shoulder motion, including elevation, angles less than 130°, and external rotation less than 45° [9]. Patients with a history of marked shoulder trauma, fracture or surgery, rotator cuff rupture, infective shoulder disease, breast cancer, and any condition that could result in shoulder pain or shoulder ROM limitations (e.g., myocardial infarction, chronic obstructive pulmonary disease, and stroke) were excluded. Those who met the eligibility criteria were invited to participate. All eligible patients signed a consent form after they understood the entire study process and agreed to take part in the study. The demographics of the participants, including age, weight, height, and the dominant hand, were collected first. The dominant hand was identified by asking the participant which hand was most often used to write and hold chopsticks.

Patients with FS were categorized according to the severity of the two clinical symptoms (shoulder pain and restricted shoulder ROM), which was based on the score of a self-reported scale, the DASH questionnaire. This questionnaire consists of 30 items, with all scores measured on a scale of 0 to 100. The higher the score, the greater the functional disability. Based on data from previous studies, those with scores less than 30 are considered to be having less severe disabilities and might be able to perform their regular duties [21,22,23]. Accordingly, a score of 30 was determined as the cut-off point for this study, and the participants were divided into two groups: the high- and low-disability groups. 

### 2.2. Procedures

All assessments and measurements of the participants in this study were taken prior to their scheduled treatment time, including measurements of shoulder pain intensity and the shoulder ROM, and the participants completed self-administered questionnaires.

Each participant was asked to self-report their worst pain intensity using a 100 mm visual analog scale (VAS) during active shoulder motions in the past week. Each participant’s passive ROM (PROM) was measured using a universal goniometer on both sides of the shoulder. All participants were measured in a supine position by a senior orthopedic physiotherapist. Considering that shoulder motion in daily activities is often complex and multi-planar, four directions of shoulder motion were measured; these included flexion, abduction, and external and internal rotation at approximately 90 degrees of shoulder abduction as described in the literature, with each shoulder PROM within the patient’s tolerable pain limits or physiological limitations [24,25]. A total of three PROM measurements were taken in each direction, with a 10 s rest interval. The rest interval between movement direction changes was about 1 min. Three measurements of the shoulder PROM in each direction were recorded and averaged, and then the four directions of shoulder PROM were summed up for further data analysis. Participants were also asked to fill out the World Health Organization Quality of Life Brief Version (WHOQoL-BREF) questionnaire to indicate the individual’s self-perceived health [26]. The WHOQoL-BREF questionnaire has been translated into a Taiwanese version with cultural adaptations. It has been shown to have good reliability and validity [27]. This tool consists of 28 questions in 4 domains (physical, psychological, social, and environmental), with scores ranging from 4 to 20 in each domain. Higher scores indicate better quality of life.

### 2.3. Data Analysis

The data were recorded and analyzed using the Statistical Package for Social Sciences (IBM SPSS 20.0). Qualitative variables were expressed as absolute numbers, whereas quantitative variables were expressed as mean ± standard deviation (or median and range).

For the quantitative variables, the Shapiro–Wilk test was first used to assess whether the data were normally distributed. Since the quantitative variables used herein were not normally distributed, nonparametric statistics were used for data analysis.

For comparison between the two groups, quantitative variables (age, body mass index, duration of FS, duration of treatment, VAS, shoulder PROM, and WHOQoL score) were analyzed using the Mann–Whitney U test. Categorical variables (sex and no. of the affected side) were examined using Fisher Exact and Pearson chi-square tests. 

The relationship between WHOQoL score (physical, psychological, social, and environmental dimensions) and the two clinical symptoms (shoulder pain intensity and shoulder PROM) was examined by Spearman’s rank correlation analysis, performed separately for the high- and low-disability groups. The level of statistical significance for all analyses was set at 0.05.

## 3. Results

A total of 49 patients with unilateral FS participated in this study, their mean age was 55.1 ± 8.4 years, and more than half of them were female and had the non-dominant shoulder as the affected side. Those patients with FS had significant shoulder pain intensity and restricted shoulder PROM. This study was based on a 30-point DASH score cut-off, with 29 patients in the high-disability group (mean DASH score of 51.9 ± 13.7) and 20 in the low-disability group (mean DASH score of 20.4 ± 7.1). The demographic and clinical characteristics and the WHOQoL score of the two groups are summarized in Table 1. There were no significant differences between the two groups in terms of demographic characteristics and duration of FS. The high-disability group had lower WHOQoL scores than the low-disability group in the four dimensions. There was a significant difference between the two groups in WHOQoL scores for physical, psychological, and social dimensions (but not for the environmental dimension) (Table 1) (*p* < 0.05).

### 3.1. Correlations between Clinical Symptoms and the WHOQoL Score

For both the high- and low-disability groups of patients with FS, no significant correlations were found between clinical symptoms (pain intensity and shoulder PROM) and WHOQoL scores (physical, psychological, social, and environmental dimensions) (Table 2).

### 3.2. Correlations of WHOQoL Scores between Four Dimensions

In Table 3, in the high-disability group, there was a significant inter-dimensional correlation between the four dimensions of the WHOQoL scores, showing a high-to-moderate positive correlation, with rho correlation coefficients ranging from 0.60 to 0.74 (*p* < 0.01).

In the low-disability group, only the psychological and social dimensions and the psychological and environmental dimensions had significant positive correlations (rho correlation coefficients of 0.50 (*p* < 0.05) and 0.60 (*p* < 0.01), respectively), whereas the other dimensions did not have significant correlations.

## 4. Discussion

The patients with FS participating in this study had significant shoulder pain and restricted shoulder motion. In our study, the DASH scale was used to assess the severity of the shoulder pain and restricted shoulder motion, with a score of 30 points as a cut-off point to distinguish between high- and low-disability groups. From the demographic data, the age distribution of the patients in both groups was middle-aged. The other data parameters, including the ratio of sexes, the proportion of FS occurring on the left or right shoulder, and the duration of FS, were similar in the two groups. The differences in the WHOQoL-BREF questionnaire scores of the two groups of patients were compared in order to display the conditions of self-perceived health in, respectively, the patients with severe and mild FS symptoms. 

Previous research on self-perceived health in patients with FS has mainly discussed the physical and psychological aspects of health [8,28]. In our study, two additional health aspects were included to provide a more comprehensive profile of living healthily in patients with FS. This analysis is not only intended to increase understanding of the patient’s self-perceived health, but also to highlight the important concept that consideration of patients’ self-perception of health is required for multifaceted and diverse professional care.

Are there any norms or standardized data that can be used for comparing scores on the WHOQoL questionnaire? As far as we currently know, no normative WHOQoL score has been found specifically for patients with FS. In this study, by comparing the WHOQoL scores of the two groups, the scores of the low-disability group could help reflect the meaning and problem level of self-perceived health in the high-disability group. The results of this study also offer possible clinical applications, in that healthcare professionals could use changes in the WHOQoL score to understand how a patient’s health is improving over time or use the score as an indicator to confirm the degree of progression during therapeutic intervention. 

In clinical practice, healthcare professionals often focus on relieving pain and improving motion in patients with FS, which seems to imply that improving these obvious symptoms is the sole goal of professional intervention [13,29]. Indeed, the treatment of FS usually requires a long course of healthcare that may gradually influence the functioning of a patient’s whole daily life [4,30]. This study analyzed the correlation between clinical symptoms of FS and patients’ self-perceived health, in order to verify whether the degree of clinical symptoms correlates with health-related quality of life. If so, reducing pain and increasing ROM will help improve the patient’s perceived health and well-being. From the results of this study, it was found that there was no significant correlation between the clinical symptoms and WHOQoL scores of the two groups of patients; in other words, whether the clinical symptoms were severe or not had no correlation with patients’ self-perceived health, which was different from our expected results. It is suggested that, in the healthcare of patients with FS, therapeutic intervention should not only focus on pain relief and improvement of the shoulder ROM but should also include therapeutic training oriented to the patient’s life situation. For example, how to overcome the difficulty of using the upper limb high over the shoulder, when dressing in daily life, and how to effectively use both upper extremities in situations that social interaction in life needs, without experiencing serious obstacles. If healthcare professionals pay attention to these function-oriented and task-specific needs of a patient during treatment, they will improve the patient’s experience of healthy living. Therefore, it is suggested that the findings of this study can be applied to both clinical practice (i.e., focusing on both the patient’s symptoms and the patient’s daily health) and healthcare policy (i.e., the concept of patient-centered care).

In this study, another perspective on the self-perceived health of patients with FS is provided. There is an interactive correlation between the dimensional scores of the WHOQoL questionnaire; that is, when one of the dimensions has a better self-perceived health score, it may affect other dimensions. From this, we suggest that in clinical treatment, the effective improvement of one of the four aspects may relatively improve the health experience of other aspects; for example, improving the self-perceived environmental aspect of health may improve the physical or psychological aspects of health. It is believed that more and different research perspectives could inspire the use of appropriate intervention strategies when dealing with the problems of patients with FS in clinical practice.

There are some limitations that need to be noted in this study. First, because the patients in this study were based in local areas, a number of samples from multiple areas may be required to extrapolate to the general population. Secondly, this is a cross-sectional study, and further longitudinal studies or post-intervention follow-up may be needed to examine the temporal relationship between changes in FS patients’ self-perceptions of health and their clinical symptoms. Thirdly, the patients with FS recruited in this study suffered from unilateral FS and most of them had the non-dominant limb as the affected side. In order to ascertain whether the correlation between clinical symptoms and the patient’s self-perceived health is different from the present results when the dominant limb is the affected side, or when FS is bilateral FS, further research is required. Furthermore, the patients in this study were middle-aged; whether our study results can be applied to patients of other age groups needs to be explored by future research. 

## 5. Conclusions

This study revealed that the self-perceived health of patients with FS is independent of their shoulder pain intensity and their degree of restricted shoulder motion. Patients with FS who have severe clinical symptoms experience a wider range of impacts on their self-perceived health. Therefore, it is suggested that healthcare providers should pay more attention to the self-perceived health needs of FS patients while addressing their clinical symptoms.

## Figures and Tables

**Table 1 ijerph-19-14396-t001:** Demographic characteristics, clinical symptoms, and the WHOQoL scores in the high- and low-disability groups.

	Groups	High-Disability (*n* = 29)	Low-Disability (*n* = 20)	
Variables		Mean ± SD	Mean ± SD	*p* Value
Age (years)		55.5 ± 8.0	54.6 ± 9.1	0.887
BMI (kg/m^2^)		24.8 ± 4.5	23.7 ± 3.5	0.339
Sex, no. (%)				0.221
Male		11 (22.4)	4 (8.2)	
Female		18 (36.7)	16 (32.7)	
Affected side, no. (%)				0.690
Dominant		10 (20.4)	8 (16.3)	
Non-dominant		19 (38.8)	12 (24.5)	
Duration of FS (weeks)				0.597
Median		17.4	20.1	
Range		6.0–257.7	6.7–271.4	
Duration of Treatment (weeks)				0.610
Median		0	1	
Range		0–28	0–20	
Pain intensity (VAS_mm_)		79.9 ± 18.1	55.6 ± 23.8	0.000 **
Shoulder PROM (degrees)				0.005 **
Sound side		489.5 ± 67.5	513.0 ± 23.7	
Affected side		246.3 ± 60.5	297.5 ± 67.7	
WHOQoL scores				
Physical		12.7 ± 2.8	14.0 ± 1.1	0.044 *
Psychological		12.7 ± 2.8	14.2 ± 1.5	0.029 *
Social		13.3 ± 2.6	14.3 ± 1.5	0.029 *
Environmental		13.5 ± 2.8	14.8 ± 1.0	0.074

* *p* < 0.05; ** *p* < 0.01. BMI, body mass index.

**Table 2 ijerph-19-14396-t002:** Correlation coefficients between clinical symptoms and WHOQoL scores.

Groups	High-Disability (*n* = 29)		Low-Disability (*n* = 20)	
WHOQoL Scores	Pain Intensity	Shoulder PROM	Pain Intensity	Shoulder PROM
Physical	−0.18	−0.07	0.22	−0.41
Psychological	−0.00	−0.12	−0.05	−0.18
Social	0.17	−0.22	0.06	0.07
Environmental	0.01	−0.17	−0.08	−0.08

**Table 3 ijerph-19-14396-t003:** Correlation coefficients of WHOQoL scores between the four dimensions.

Groups	High-Disability (*n* = 29)				Low-Disability (*n* = 20)			
WHOQoL Scores	Physical	Psychological	Social	Environmental	Physical	Psychological	Social	Environmental
Physical		0.73 **	0.60 **	0.64 **		0.27	0.30	0.26
Psychological			0.72 **	0.74 **			0.50 *	0.60 **
Social				0.71 **				0.34

* *p* < 0.05; ** *p* < 0.01.

## Data Availability

The data presented in this study are available on request from the corresponding author.

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
