# Peer review of "The Relationship between Self-Perceived Health and Clinical Symptoms in Patients with Frozen Shoulders"

_ijerph, 2022, doi:10.3390/ijerph192114396_

Round 1

Reviewer 1 Report

Thank you for sharing this article with me. This is very well written and I have only minor comments for the authors.

1. I see that the authors point to the importance of self-perceived health needs as a result of the significant relationship between pain and self-rated health. While your point and valid and important, isn't it also possible that pain may be representing a part of this multidimensional nature of self-rated health? If so, please discuss this in the paper.

2. Maybe it is useful to discuss the cross-sectional nature of this study as part of the limitation? We cannot test the temporal order.

3. I like your clinical suggestions. Can you integrate these suggestions within the context of existing health policy?

Reviewer 2 Report

Manuscripts establish the relation between shoulder mobility and severity of pain with health perception/quality of life in people with frozen shoulder.

 Line 18-31: Please remove abbreviations from the abstract.

-          Line 34: Introduction. More information about the cause of frozen shoulder should be added to contextualize the background of this pathology.

-          Line 80: data are quite old…

-          Line 80: “30-70 years” have you considered to compare different range of years?

-          Line 87: have you considered the origin of the frozen shoulder as an inclusion criteria?

-          Line 94: if you explain how have you collected the demographic data, you should explain the others too (i.e. a self-administer questionnaire, asking the patient…)

-          Line 100: “zero to 100”: please follow the same format. You can write numbers with numbers or could be written with letters but the same format should be used.

-          Line 111: the ROM was collected once or did you do these movements more times and take the media?

-          Line 137: is there any blindness in the study?

-          A better explanation about the procedure should be added (when, how, which order…)

-          Line 138: results: is there any participant receiving treatment from a specialist to solve this frozen shoulder? If the answer is positive, have you collected this data? From whom the treatment has been implemented? Since them or how long? Also, pharmacological data should be collected too to avoid bias.

-          Line 172: discussion: “cut off of 30” I have seen the reference you written but the reason which justify this decision should be explained.

-          Line 210: have you considered to use some functional scale to better reflects the daily limitations these patients had?

-          Line 229: some information about treatment and pathology duration should be mention and compare with literature.

-          Line 238: limitations, what about genre as a limitation to collect sample? The lack of physical or pharmacological treatment as limitations should be added.

-          Some mention about the multidimensional treatment these patients needs should be added.

Round 2

Reviewer 2 Report

Dear authors, 

Thank you very much for taking into account my considerations. Manuscript looks better.

Good job